# Brassinosteroid Signaling Downstream Suppressor BIN2 Interacts with SLFRIGIDA-LIKE to Induce Early Flowering in Tomato

**DOI:** 10.3390/ijms231911264

**Published:** 2022-09-24

**Authors:** Maqsood Khan, Bote Luo, Miaomiao Hu, Shangtan Fu, Jianwei Liu, Meng Jiang, Yan Zhao, Shuhua Huang, Shufen Wang, Xiaofeng Wang

**Affiliations:** State Key Laboratory of Crop Stress Biology in Arid Areas, College of Horticulture, Northwest A&F University, Yangling, Xianyang 712100, China

**Keywords:** tomato, SlFRLs, BIN2, flowering, BR signaling

## Abstract

Brassinosteroid (BR) signaling is very important in plant developmental processes. Its various components interact to form a signaling cascade. These components are widely studied in Arabidopsis; however, very little information is available on tomatoes. Brassinosteroid Insensitive 2 (BIN2), the downstream suppressor of BR signaling, plays a critical role in BR signal pathway, while FRIGIDA as a key suppressor of Flowering Locus C with overexpression could cause early flowering; however, how the BR signaling regulates FRIGIDA homologous protein to adjust flowering time is still unknown. This study identified 12 FRIGIDA-LIKE proteins with a conserved FRIGIDA domain in tomatoes. Yeast two-hybrid and BiFC confirmed that SlBIN2 interacts with 4 SlFRLs, which are sub-cellularly localized in the nucleus. Tissue-specific expression of *SlFRLs* was observed highly in young roots and flowers. Biological results revealed that SlFRLs interact with SlBIN2 to regulate early flowering. Further, the mRNA level of *SlBIN2* also increased in *SlFRL-*overexpressed lines. The relative expression of *SlCPD* increased upon *SlFRL* silencing, while *SlDWF* and *SlBIN2* were decreased, both of which are important for BR signaling. Our research firstly provides molecular evidence that BRs regulate tomato flowering through the interaction between SlFRLs and SlBIN2. This study will promote the understanding of the specific pathway essential for floral regulation.

## 1. Introduction

Floral transition is the evolution from the vegetative phase to the reproductive phase in flowering plants. This transition is accurately organized by the plant at an appropriate time to ensure reproduction. Different flowering genetic pathways were studied in *Arabidopsis thaliana*, a model plant, to integrate various flowering signals, ensuring proper timing for floral transition [1,2]. Flowering Locus T (FT) is a major florigen found in *Arabidopsis thaliana*, synthesized mainly in leaf phloem. It is localized by FT-interacting protein 1 (FTIP1) and several others through the endoplasmic reticulum membrane and transported to shoot apical meristem [3,4]. FRIGIDA (FRI) is a key protein in *A. thaliana* that regulates floral transition by activating the Flowering Locus C (FLC) [5]. It performs like a scaffold protein that interacts with other proteins and accumulates a complex, binding to the *FLC* promoter region, triggering its expression and subsequently inhibiting flowering. FRLs (FRIGIDA-Like genes) have also been reported in many sequenced plant genomes [6]. It is mainly associated with flowering regulation; furthermore, genes of this family are also involved in other biological processes that are related to reproduction, including embryo development and seed maturity [6,7]. In winter Arabidopsis ecotypes, *FLC* expression is increased by plant-specific protein FRIGIDA (FRI), binding to FRIGIDA LIKE 1 (FRL1), *FLC* expressing factor (FLX), (SUF4), and FRIGIDA ESSENTIAL 1 (FES1), resulting in a stable core protein complex [8]. In this complex, the SUF4 subunit is a DNA-binding protein that plays a role in the recruitment of FRIc to the FLC site [8,9]. The expression of *FLC* is inhibited by prolonged cold conditions, which allows the plant to bloom in the next spring [10]. Upregulation of *FRIc*-dependent *FLC* requires several chromatin modifiers, transcriptional co-activators, and a pre-transcriptional mRNA processor [11]. In the presence of functional *FRI*, COMPASS-like is enriched in *FLC* chromatin, and active *H3K4* trimethylation (H3K4me3) accumulates to promote *FLC* expression [12]. Furthermore, FRI directly interacts with *EFS* to promote *H3K36* trimethylation (H3K36me3) on *FLC* [13].

BIN2*,* a key suppressor of BR signaling, plays an important role in signaling transmission. Researchers have progressively shaped its complex regulatory mechanism. Phosphorylation occurs as a result of BR binding to the BRI1/BAK1 complex and activates plasma membrane-localized BSK and CDG, which, upon activation of BRI1 inhibitory factor 1 *(*BSU1), dephosphorylate BIN2 [14,15]. During early BR signaling, BIN2 rapidly degrades through the 26S proteasome [16]. Due to this degradation, the interaction between BIN2 and substrates is blocked [17]. To avoid the inhibitory effects of BIN2 on BZR1 and BES1 in the nucleus, BRs induce the plasma membrane localization of BIN2 through OCTOPUS (OPS) [18]. However, it is mainly expressed in the phloem, due to which BIN2 regulation through OPS is limited to the phloem and promotes its differentiation. Additionally, HSP90, upon interaction with BIN2, keeps it in the nucleus. BRs advance the complex of BIN2-HSP90 to the cytoplasm, enabling BR signaling export [19]. Recent studies showed that BIN2 regulates growth and development in plants. It is involved in the regulation of stomata and also has an important role in weakening MAPK signals [20]. It reduces the activity of CSC protein; as a result, BES1/BZR1 regulates the expression of *CESA1* and its homologous genes, and co-regulation of BIN2 and BES1/BZR1 induces cell elongation during growth [21]. The regulatory mechanism of BIN2 is very complex. It also could phosphorylate many important substrates in different signaling pathways [22]. Due to its role as a multifunctioning kinase, BIN2 is essential in plant processes. 

BR is involved in a complex mechanism to promote floral transition. Synthetic mutant of BR shows slow leaf differentiation and delayed flowering, suggesting that it promotes flowering [23]. In Arabidopsis, *FLC* and its homologues are repressed in BR mutants, which suggests that BR antagonizes the autonomic pathway by constitutively activating *FLC* expression during vegetative growth [24]. The tomato (*Solanum lycopersicum* L.) is an important vegetable crop cultivated on 5 millions hectares, with production of over 180 millions tons [25]. It is native to South America and is famous for its unique flavor. It is a rich source of nutrients and is especially a good source of carotenoids and vitamin C, which makes it an important vegetable crop [25]. Additionally, it has a simple genome, which enables researchers to use it as a model plant for breeding research. This research was designed to identify interactions among different FRLs and BIN2. Furthermore, different mutants were used to explore the impact of SlFRIGIDA-LIKE and SlBIN2 on the flowering time of tomatoes. 

## 2. Results

### 2.1. Cluster Analysis of Tomato FRIGIDA Gene Family

A component interacting with SlBIN2 (*SlBIN2:Soly02g072300*) was obtained from a yeast two-hybrid screening library. After sequencing, it was searched in the tomato database (https://solgenomics.net/; accessed on 11 November 2015), and it was found that the component resembled the *Solyc09g076050* gene. The blast alignment of this protein sequence in the Arabidopsis data library (https://www.arabidopsis.org) shows that it is homologous to the *At4g14900* gene. It was identified that *At4g14900* gene encodes a member of the Arabidopsis FRIGIDA-LIKE protein family-AtFRL4. In Arabidopsis, AtFRL4 regulates early flowering, while its overexpressed lines show a late-flowering phenotype [26].

To identify the interactive effect of Solyc09g076050 protein with SlBIN2 and FRIGIDA-LIKE proteins in tomatoes, we have considered 12 proteins that contain the FRIGIDA domain, including Solyc09g076050, Solyc11g071340, Solyc06g007640, Solyc06g082930, Solyc04g008840, Solyc01g098240, Solyc12g042440, Solyc06g074450, Solyc04g072200, Solyc03g117910, Solyc03g117920, and Solyc03g117930 in the tomato latest database (SL4.0 and ITAG4.0 versions), named SlFRL1-12. Through protein conserved domain analysis, we have found that 10 of them (SlFRL1-9 and SlFRL12) have conserved domain and conserved sites, while the other 2 members with FRIGIDA-like proteins (SlFRL10 and SlFRL11) have less conserved domains. More interestingly, SlFRL6 contains two FRIGIDA domains; one of them is highly conserved. Additionally, SlFRL3 has a relatively conserved outer membrane protein OmpH-like domain near the N-terminus. Then, the amino acid sequences of FRIGIDA homologous proteins in tomato and Arabidopsis were downloaded, and MEGA8 software was used to create a phylogenetic tree, as shown in Figure 1. The 12 SlFRLs proteins in tomato are located in three subfamily branches, among which SlFRL3-5 are located in the third clade, SlFRL1-2 are located in the fourth clade, and the rest are located in the first clade.

### 2.2. Yeast Two-Hybrid Assay to Screen the Interaction between SlFRLs and SlBIN2

To confirm whether SlFRLs interact with SlBIN2, we performed a yeast two-hybrid assay. The functional genomic database (http://ted.bti.cornell.edu/cgi-bin/TFGD/digital/home.cgi; accessed on 16 January 2016) was used to find the site of *SlFRLs*, and they were then extracted from the point with the highest expression. Eight of them (*SlFRL*1, 2, 3, 4, 5, 7, 8, and 9) were cloned successfully. All of the 8 *SlFRL* CDS sequences are recombined into a pGADT7 vector, double digested by *Eco*R I/*Bam*H I, fused with the BD-SlBIN2 vector and transformed into a yeast cell AH109. The cells were cultured at 30 °C on a four-deficient solid medium (SD/-Ade-Leu-Trp-His) for four days, and then the healthy colonies were selected and cultured for a further four days. Results (Figure 2) indicated that AD-SlFRLs and BD-SlBIN2, positive controls AD-SlBZR2 and BD-SlBIN2, and negative controls AD-SlFRLs and BD-Empty have normal growth on two-deficient medium, indicating that each fusion yeast strain grows well. However, AD-SlFRL1 and BD-SlBIN2, AD-SlFRL3 and BD-SlBIN2, AD-SlFRL5 and BD-SlBIN2, AD-SlFRL9 and BD-SlBIN2 were found on four-deficient medium and color indicator medium. SlBIN2 and the positive control can grow normally and show blue color on solid medium containing X-α-gal. Results were stable and proved that SlFRL1, SlFRL3, SlFRL5, and SlFRL9 could interact with SlBIN2 in vitro.

### 2.3. Bimolecular Fluorescence Complementation Assay Confirm That *SlFRLs* Physically Interact with *SlBIN2*

Results of the yeast two-hybrid assay proved that four members, SlFRL1, SlFRL3, SlFRL5, and SlFRL9, can interact with SlBIN2 in vitro. To further investigate whether these four SlFRLs interact with SlBIN2 in vivo, these four *SlFRLs* and *SlBIN2* were constructed in pER8-NeYFP and pER8-CeYFP and after the transformation of Agrobacterium, with the helper strain P19, and pBI121-35S-Nls-DsRed (labeling the nucleus under red fluorescence excitation) strains according to 1:1:1:1. To induce expression, 1 μM β-estradiol was sprayed each day, and after three days, confocal microscopic observation was carried out. According to the results (Figure 3), upon excitation of yellow fluorescence, it can be seen that the combination of SlFRL-NeYFP and SlBIN2-CeYFP has nuclear-like fluorescence. Similarly, the positive controls SlBZR1-NeYFP and SlBIN2 and the combination of CeYFP showed fluorescence in the nucleus and cell membrane. On the other hand, no fluorescence was found in negative controls, which indicated that BIN2 interacts with all four SlFRLs in tobacco leaf epidermal cells. It can also be noticed that under red fluorescence, upon excitation, the combination of SlFRL-NeYFP and SlBIN2-CeYFP appears to demonstrate red fluorescence. It can be seen that the fluorescence is located in the nucleus due to the nuclear localization feature of nuclear localization signals. These results together with the results of the yeast two-hybrid assay demonstrated that these four SlFRLs can interact with SlBIN2 in vitro and in vivo.

### 2.4. SlFRL Expression Analysis and Subcellular Localization

To explore the expression of *SlFRLs* in different plant parts, total RNA from various parts, such as stems, young leaves, flowers, young and old roots, ripe green fruits, and red ripe fruits of tomato MT-d^+^, was extracted from 8-week-old seedlings. It was reverse transcribed in cDNA and analyzed by quantitative fluorescence PCR. Results (Figure 4) demonstrated that *SlFRL1* is highly expressed in young roots and flowers; the lowest expression of *SlFRL1* was noticed in green fruits. *SlFRL3’s* higher expression was similar to *SlFRL1,* while lower *SlFRL3* was found in mature fruits. *SlFRL5* has a different pattern of expression, as the highest expression was noticed in fruits, while lower expression was observed in old roots. In the case of *SlFRL9*, young roots have the highest expression, followed by flower and green fruit, while the lowest was found in mature fruits. Overall results show that *SlFRLs* are highly expressed in flowers and young roots.

Previous studies showed that AtBIN2 is localized in the cytoplasm and nucleus, and AtFRI is localized in the nucleus. However, SlFRLs were still unknown where they localized. To verify SlFRLs’ subcellular localization, the tobacco leaves were injected with activated (OD_600_ = 0.5) mixed strain (35S::*SlFRLs*-GFP + 35S::NLS-DsRed + P19) and GFP empty as a positive control, while H_2_O served as a negative control. According to the results shown in Figure 5, all the four SlFRLs (1, 3, 5 & 9) showed nuclear fluorescence upon excitation of green fluorescence. While under excitation of red fluorescence, nuclear fluorescence also appeared. The nuclear fluorescence under GFP all coincided with those under RFP, which proved that the four SlFRLs were all located in the nucleus.

### 2.5. The Expression Level Comparison of the Key Components of BR Pathway in Overexpression Lines of SlFRL5 and SlFRL9

Two basic components of BR synthesis are *SlCPD* and *SlDWF.* Increases in their level will reflect the decrease in BR signaling strength for the plant signaling feedback regulation. Similarly, SlBIN2 is a core negative regulator of BR signaling, and increases in its level will inhibit BR signaling. The mRNA levels of *SlCPD*, *SlDWF* and *SlBIN2* were determined in *SlFRL5* and *SlFRL9* overexpression lines and wild type lines. The results in Figure 6 suggested that the mRNA level of *SlBIN2*, *SlCPD,* and *SlDWF* in *SlFRL5* and *SlFRL9* overexpressed lines were significantly higher than that in wild type lines. These results indicated that *SlFRL5* and *SlFRL9* reduced BR signaling levels in tomatoes.

### 2.6. Overexpression of SlFRL5 and SlFRL9 Improved Plant Height and Early Flowering in Tomato

After genetic transformation, three stable lines of *SlFRLs* T1 generation were selected and grown for 4 weeks. The results are presented in Figure 7, while A, C, E, and G are the phenotypes of *SlFRL*1, 3, 5, and 9 overexpression lines. RT-PCR was carried out to confirm the material. The plant heights of overexpressed lines of *SlFRL*1, 3, 5, and 9 are shown in B, D, F, and H. According to the results, *SlFRL5-* and *SlFRL9*-overexpressed lines have increased plant height as compared to other *SlFRLs,* which are not significantly different. 

In Arabidopsis, FRIGIDA and its homologous proteins promote the expression of FLC, which inhibits FT expression, leading to the promotion of late flowering. Late and early flowering was determined based on counting the number of true leaves under the first panicle in tomatoes. We also use the same method for *SlFRL* overexpression material and wild type plants [27]. The results shown in Figure 8 indicated that plants of *SlFRL5-* and *SlFRL9-*overexpressed lines had a decreased flowering node position. This suggested that *SlFRL5* and *SlFRL9* promote early flowering in tomato. 

### 2.7. Effects of Transient Silencing of SlFRLs on Tomato Growth and Flowering

Virus-mediated gene silencing (*VIGS*) was used to obtain transiently silenced *SlFRL* tomato plants. Albino type leaves were found in TRV2-*SlPDS* positive plants, indicating that the VIGS silencing test for *SlPDS* was successful. TRV2-*SlFRLs* transiently transformed plants (Figure 9B), and TRV2-SlFRL4X is a mixed infection of four TRV-*SlFRL* strains. To confirm silencing, we performed RT-qPCR. Reduced mRNA levels were observed in each silencing line, indicating transient silencing of *SlFRLs* (Figure 9B–D). After growing these lines for four weeks, plant height was measured for each line. It was found that the plant height was increased in all silenced lines as compared to negative controls. Additionally, the number of true leaves was counted after the first panicle at flower blooming. However, no significant difference was noticed in flowering time. The results indicated that silencing of *SlFRLs* in tomato plants increased plant height, but flowering time was not affected by *SlFRLs* silencing. Furthermore, mRNA levels of *SlBIN2*, *SlCPD*, and *SlDWF* were also determined in these silenced lines (Figure 9A). In these lines, the transcription level of *SlCPD* was increased significantly, while the transcription level of *SlBIN2* and *SlDWF* was decreased (Figure 10). 

## 3. Discussion

Current advances in plant genomics allow us to isolate different genes and find the interaction of proteins. In many plants, genes are the homologues of those reported in the model plant Arabidopsis [27]. Some of them are orthologous genes found in Arabidopsis and have different functions, while many of them are not found in Arabidopsis and have distinct functions. Plants have different types of proteins that interact with each other in various signaling pathways [28]. Yeast two-hybrid screening was carried out to identify the interaction between SlFRLs and SlBIN2. A tomato database was used for the amino acid sequence of SlFRLs. These SlFRLs were named SlFRL1-SlFRL12 according to their similarity and closeness with SlFRL1. Cluster analysis of these SlFRLs showed that they are arranged in three subfamily clades, and 7 FRLs of Arabidopsis were located in 5 different clades, while the components located in clade 2 are missing in tomato. These components of clade 2 of Arabidopsis are AtFRL1 and AtFRL2. Previous studies have shown that these two proteins are located in different ecotypes and can interact with AtFRI [29]. Studies suggested that loss of AtFRL1 and AtFRL2 leads to a loss of FRI function, resulting in early flowering phenotypes [30,31]. This suggests that the function of FRIGIDA in tomatoes might not be affected by the clade II component. Afterward, 8 genes, *SlFRL1-SlFRL5,* and *SlFRL7-SlFRL9* were cloned successfully; four of proteins SlFRL1, SlFRL3, SlFRL5 and SlFRL9 were screened through a yeast two-hybrid assay to identify whether they interact with SlBIN2 or not. Our results indicated that SlFRLs interact with SlBIN2 both in vivo and in vitro. Furthermore, subcellular localization was carried out and proved that all four genes, SlFRL1, SlFRL3, SlFRL5, and SlFRL9*,* are localized in the nucleus, similar to the results found in Arabidopsis, which are also localized in the nucleus [28]. Tissue expression analysis showed that *SlFRL1, SlFRL3, SlFRL5, and SlFRL9* had high expression levels in young roots and flowers, while relatively low expression was found in fruits. These results are consistent with the previous results reported in Arabidopsis, which showed that FRI was highly expressed in roots [32]. 

Further, we explored the biological function of SlFRLs. For this purpose, *SlFRL*-overexpressed lines were created in tomato plants. It was observed that plant height was increased in *SlFRL5*-OE and *SlFRL9*-OE lines when investigating four-week-old seedlings. Additionally, the number of true leaves was counted under the panicle of the flower to study its effects on flowering time. It was observed that both *SlFRL5*-OE and *SlFRL9*-OE bloom earlier than the wild type. *FLC* expression decreases due to the absence of FRI, while its overexpression does not increase the expression of *FLC* [33]. However, another study suggested that a late flowering phenotype was observed due to overexpression of the *PtFRI*-L gene in Arabidopsis [34]. A late flowering phenotype was found in Arabidopsis due to overexpression of *EjFRI* [27]. In Chinese cabbage, the expression levels of *BrFRIa* and *BrFRIb* are not correlated with flowering time, but overexpression of *BrFRIb* in Arabidopsis causes delayed flowering [35]. Similarly, in tobacco, the expression of *CpFRI* has no effect on flowering, but its overexpression in Arabidopsis causes inhibition of flowering. Previous studies suggested that FRIGIDA has different functions in different plants [36]. 

We further created TRV2-*SlFRL* transiently silenced plants by using VIGS. We noticed that plant height increased, and flowering time was not affected by the silencing of *SlFRLs*. This might be due to the absence of Clade II FRIGIDA in tomatoes. Additionally, silencing of *SlFRLs* may not significantly affect flowering time in tomatoes. We noticed that *SlFRL5-* and *SlFRL9-*overexpressed as well as -silenced lines have taller plants as compared to others. This may be due to the influence of homeostatic factors [37]. Both the overexpressed and knockout line of *SlBIN2* inhibit the growth and development of tomatoes, as SlBIN2 interact with SlFRLs, so the phenotype at the seedling stage might be affected by *SlBIN2* [38].

Variation in plant height of *SlBIN2-*overexpressed materials might be due to the abnormal *SlDWF*, as *SlBIN2* overexpression [39]. This is caused by the hybrid background of MT and MT-d^+^. Multi-generation backcross between MT::*SlBIN2-OE* and MT-d^+^ will inevitably lead to inconsistency in the expression level of the *DWARF* gene.

We also measured the mRNA level of *SlFRL1*, *SlFRL3*, *SlFRL5*, and *SlFRL9* in the *SlBIN2*-overexpressed and -silenced lines. We found that the expression of *SlFRL1*, *SlFRL3*, *SlFRL5*, and *SlFRL9* was negatively regulated by SlBIN2. Additionally, the mRNA levels of *SlBIN2* were measured in *SlFRL-*overexpressed and -silenced plants. *SlBIN2* expression was positively regulated by SlFRLs, as it is an important protein kinase and negative regulator; it mainly functions as a phosphorylating substrate [40]. SlBIN2 may reduce the activity of SlFRLs by phosphorylating SlFRLs. Subsequently, whether SlBIN2 protein kinase will phosphorylate SlFRLs can be detected by an in vitro phosphorylation method. We have seen that *SlBIN2* overexpression decreased the mRNA level of *SlFRLs*, while an increase in the mRNA level of *SlBIN2* was noticed in *SlFRL-*overexpressed lines. This might be due to the role of SlBIN2, which can reduce the translation level of downstream genes by inhibiting the activity of BES1/BZR1 transcription factors [41]. In addition, both overexpression and knockout of *SlBIN2* in tomatoes will reduce plant height, which has the characteristics of steady-state regulation of growth and development [38]. We assume that there may be another pathway between these two (SlBIN2 and SlFRLs), and there may be other transcriptional regulators involved among them. Overexpression of *SlBIN2* may inhibit the activity of this intermediate transcriptional factor and, as a result, reduce the translation of *SlFRLs*, as SlBIN2 has the ability to reduce the translation of downstream genes through inhibiting BES1/BZR1 [42]. Still, we do not have any perfect explanation for the overexpression of *SlFRLs* increasing *SlBIN2* mRNA levels. There might be some other factors involved in this process that need to be explored further. 

The flowering time is closely related to the fruit maturity shifting to an earlier date and the total yield of tomatoes. It is important to study the mechanism of regulation of flowering time for the breeding and production of tomatoes [43]. BRs play multiple critical roles in plant growth and development, and the BR signal pathway is the main target pathway of crop molecular breeding at present. The research on the main negative regulatory component BIN2 of this path and downstream components FRLs regulating flowering has important practical significance for molecular tomato breeding in regulating flowering time to obtain high-quality tomato cultivars [44,45].

## 4. Materials and Methods

### 4.1. Plant Material and Culture Conditions

The tomato MicroTom-d^+^ (MT-d^+^) was used as plant material. To reduce the *DWARF* gene mutation influence in MT, we used MT-d^+^ (LA4470) and the *DWARF* gene restoration function (MT-d^+^) stocked in the TGRC (Tomato Genetics Resource Center (ucdavis.edu). To conduct fluorescent bimolecular complementary labeling (BiFC) experiments, we used N. benthamiana (*Nicotiana tabacum* var N. benthamiana) as transient transformation material. Other cultural conditions were the same as followed by reference [46] with some modifications. Seeds were sown in small pots and kept in the dark for germination at 25 °C. After germination, they were kept at room temperature for 7 days, having 70% relative humidity and a light period of 16 h. Seedlings were irrigated frequently to avoid wilting. They were transferred to pots for experiments and kept in the greenhouse for further investigation. 

### 4.2. Strains and Vectors

In the experiment, we used the AH109 strain of *Pichia pastoris* and the GV3101 strain of Agrobacterium, which also included the DH5α strain of *Escherichia coli*. Furthermore, the vectors were preserved and used for yeast two-hybrid (Y2H) vectors pGADT7 and pGBKT7 for BiFC pER8-CeYFP and pER8-NeYFP vectors, while pGBKT7-*SlBIN2.1* and pGADT7*-SlBZR2* were used as positive control for Y2H. Cell nuclear marker strain GV3101::pBI121-35S-NLS-DsRed with helper infection strain GV3101::P19 was used in the BiFC test. For VIGS, we used the GV3101 strain of Agrobacterium and DH5α of *E. coli*. This experiment also included pBI121-35S-FLAG and pBI121-35S-GFP plant overexpression and pTRV1 and pTRV2 virus-induced gene silencing (VIGS) vectors. 

### 4.3. Gene Cloning

The Genome Database of Arabidopsis (http://www.arabidopsis.org; accessed on 16 January 2016) and Genome Database of Tomato (https://solgenomics.net/; accessed on 16 January 2016) were used to obtain FRIGIDA and its homologous proteins. To download the protein sequence of the candidate gene, the BLAST tool was used to perform homologous gene blast analysis, and MEGA8 software was used to construct a phylogenetic tree. Tomato FRIGIDA-LIKE homologous genes (*SlFRLs*) were obtained according to the distance and clustering of each clade of the phylogenetic tree. To confirm the mRNA level of *SlFRLs* in specific tissues for the highest expression, the Tomato Functional Genome Database (http://ted.bti.cornell.edu/cgi-bin/TFGD/digital/home.cgi) was used. For further processes, the protocol previously described was used [46].

### 4.4. Yeast Two-Hybrid

For yeast two-hybrid analysis, we used the method previously described [43] with some modifications. The *SlFRLs* CDS sequence was recombined with the pGADT7 vector and digested by *Eco*R I and *Bam*H I, and the *SlBIN2* CDS sequence was constructed on the pGBKT7 digested by *Eco*R I and *Bam*H I. Then, we transformed it into *Escherichia coli DH5α* and extracted it. Yeast strain AH109 was inoculated on YPDA solid medium and cultured for 3–5 days at 30 °C. A single colony was picked from the cultured yeast plates, inoculated into 3 mL of liquid YPDA medium and incubated overnight at 30 °C with shaking at 230 g. Upon growth of colonies, a single colony was picked and cultured in a sterile tube containing 1 mL of two-deficient liquid medium (–Leu/–Trp) at 30 °C with shaking at 200 g overnight. We then picked 3 colonies from each cultured plate, pipetted 5 μL of this bacterial solution into a four-deficient solid medium (–Ade/–His/–Leu/–Trp, solid medium with 2% agar powder), and kept it at 30 °C. A physical interaction exists in the combination of growth on deficient medium and control growth on four deficient media.

### 4.5. Bimolecular Fluorescence Complementation Assay

Binary vectors pER8-CeYFP and pER8-NeYFP were used to ligate *SlFRLs* and *SlBIN2* and transform them into Agrobacterium GV3101. Tobacco was cultured for 6–8 weeks, and plates were lined to activate SlFRLs-CeYFP, SlBIN2-NeYFP, pER8-CeYFP and pER8-NeYFP empty, while the positive controls were SlBZR1-CeYFP, P19 and the pBI121-35S-NLS-DsRed (NLS, nuclear localization sequence) strain. This newly activated agrobacterium was inoculated into 3 mL LB containing Spe^+^ 50 μg/mL, Gen^+^ 50 μg/mL, Rif^+^ 40 μg/mL, and cultured overnight at 200 rpm at 28 °C. On the next day, a bacterial solution of 7.5 μL was added to 5 mL of liquid LB containing Spe^+^ 50 μg/mL. After the OD value of the solution reached 0.6–1.0, the Agrobacterium was collected through centrifugation. We centrifuged this solution again after being suspended in 2 mL of induction medium (0.5% D-glucose solid, 0.076% Na_3_PO_4_·12H_2_O solid, 50 mM MES-KOH, 1 M AS) to collect bacterial fluid. The resulting pellet was suspended in induction media and adjusted to OD_600_ = 0.5 and kept at room temperature for 2–4 h. Four strains of SlFRLs-CeYFP, SlBIN2-NeYFP, P19 and pBI121-35S-NLS-DsRed were added in a ratio of 1:1:1:1 and injected with a syringe to 6–8-week-old *N. benthamiana* leaves. After infection, the cells were kept in the dark for 24 h, the membrane was removed and 1 μM β-estradiol (containing 0.1% Tween) was sprayed. The infection was injected for 2–7 days, and the results were observed by laser confocal microscopy.

### 4.6. Tissue Expression Analysis of SlFRLs

For tissue expression analysis, we followed the method used by [47]. Total RNA was extracted from young roots, old roots, stems, leaves, flowers, ripe green fruits and red ripe fruits of the same tomato MT-d^+^ and reversed into cDNA. Then, we carried out the quantitative fluorescence test, taking 200 ng of cDNA as the template and carrying out the PCR reaction on the machine. Results were calculated using 2^−ΔΔCT^.

### 4.7. Subcellular Localization of SlFRLs

The *SlFRL* CDS sequence was homologously recombined with the pBI121-35S-GFP vector double-digested by *Xba* I/*Kpn* I to transform *DH5α* and GV3101. The infection method is the same as described previously.

### 4.8. VIGS-Mediated Transient Silencing of SlFRLs

Genomic DNA of tomato MT-d^+^ was used as a template. Fragments having a size of 500 bp in the *SlFRL*s gene were cloned and recombined into the pTRV2 vector cut by *Xba* I and *Kpn* I. TRV2- *SlFRLs* were used for silencing the tomato *SlFRLs* gene. The same method was used to construct the TRV2-*SlPDS* vector for silencing the tomato *SlPDS* gene as a positive control. The pTRV2 empty vector was used as a negative control. The above-constructed vector was transformed in agrobacterium GV3101. Then, it was cultured on liquid LB medium (containing Gen^+^50 μg/mL, Rif ^+^40 μg/mL, Kan^+^ 50 μg/mL) at 28 °C and 200 rpm kept overnight. Subsequently, 100 mL was added to 50 mL liquid medium (containing Gen^+^ 50 μg/mL, Rif^+^40 μg/mL, Kan^+^ 50 μg/mL, 10 mM MES, 20 μM AS) overnight at 28 °C and 200 rpm, and bacteria were collected at 1000 rpm for 10 min. It was then suspended overnight (containing 10 mM MgCl_2_, 10 mM MES and 200 μM AS), adjusted to an OD_600_ = 1.5, and placed in the dark at room temperature for 3–4 h; GV3101:pTRV1 Mix with the above bacterial solution in a ratio of 1:1. After expanding tomato cotyledons and seedling growth, the mesophyll cells were injected with a 1 mL syringe. After 2–3 weeks, positive control TRV2-SlPDS-infected tomato seedlings appeared to have the leaf albino phenotype.

### 4.9. Statistical Methods

MS Excel was used for statistical analysis. Each experiment had at least 6 independent biological replicates. The differences between the two types of plants were statistically significant using Student’s *t*-test (* *p* < 0.05, ** *p* < 0.01, *** *p* < 0.001), and the significance of differences between plants of more than two types was analyzed using analysis of variance (ANOVA). Graphs were made using GraphPad 8.

## 5. Conclusions

Our research has identified and confirmed the presence of FRIGIDA family proteins for the first time in tomatoes. Overall results stated that SlFRLs interacted with BIN2 and influenced tomato flowering. Molecular analysis showed that SlFRLs were also involved in BR signaling feedback gene regulations. Therefore, it can be concluded that it is involved in different processes related to BRs. The interaction between SlFRLs and BIN2 confirmed in this study has made a solid basis for further investigation of the detailed molecular mechanism of BR signaling regulating tomato flowering.

## Figures and Tables

**Figure 1 ijms-23-11264-f001:**
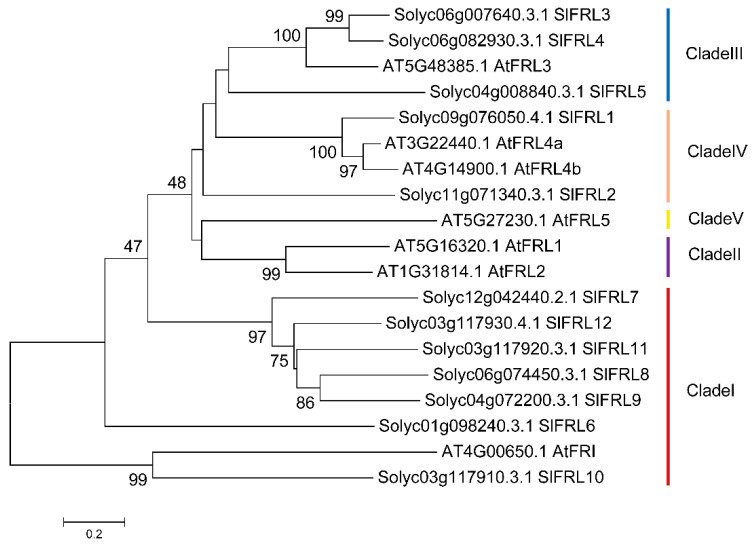
Cluster analysis of Frigida homologues in *S. lycopersicum* and *A. thaliana*.

**Figure 2 ijms-23-11264-f002:**
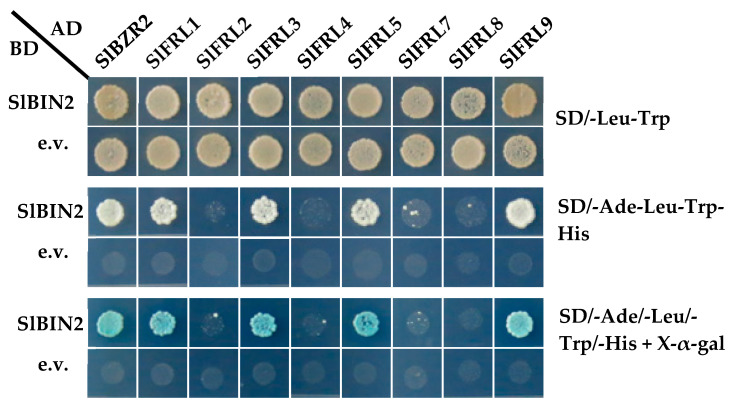
Four SlFRLs physically interact with SlBIN2 in yeast.

**Figure 3 ijms-23-11264-f003:**
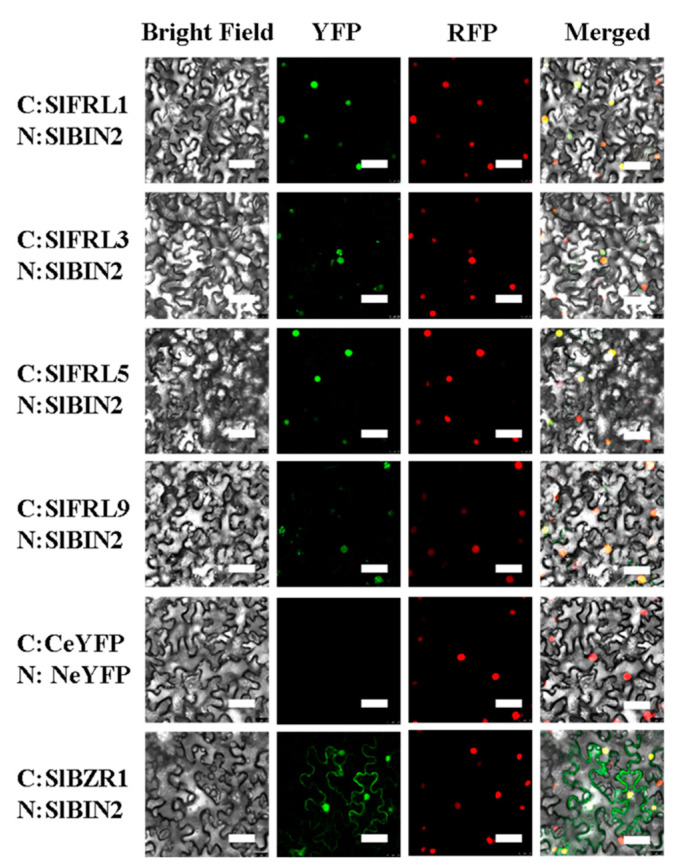
SlFRLs physically interact with SlBIN2 in mesophyll cells of tobacco (BiFC analysis). YFP is a yellow fluorescent protein; RFP is a red fluorescent protein; bar = 50 μm.

**Figure 4 ijms-23-11264-f004:**
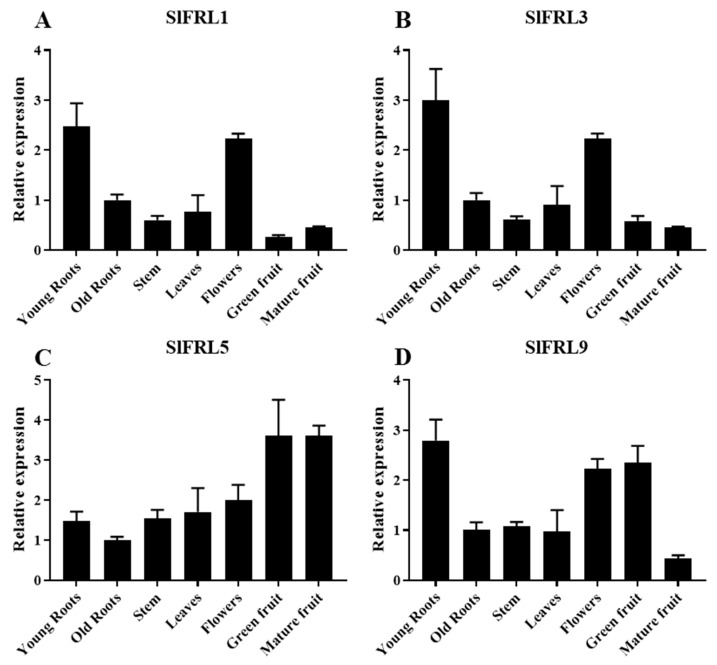
The relative expression of genes *SIFRL1* (**A**), *SIFRL3* (**B**), *SIFRL5* (**C**), *SIFRL9* (**D**) in different parts in the MT-d^+^. Three samples were expressed as mean ± standard deviation (error line). *SlUBI3* was used as the internal reference gene.

**Figure 5 ijms-23-11264-f005:**
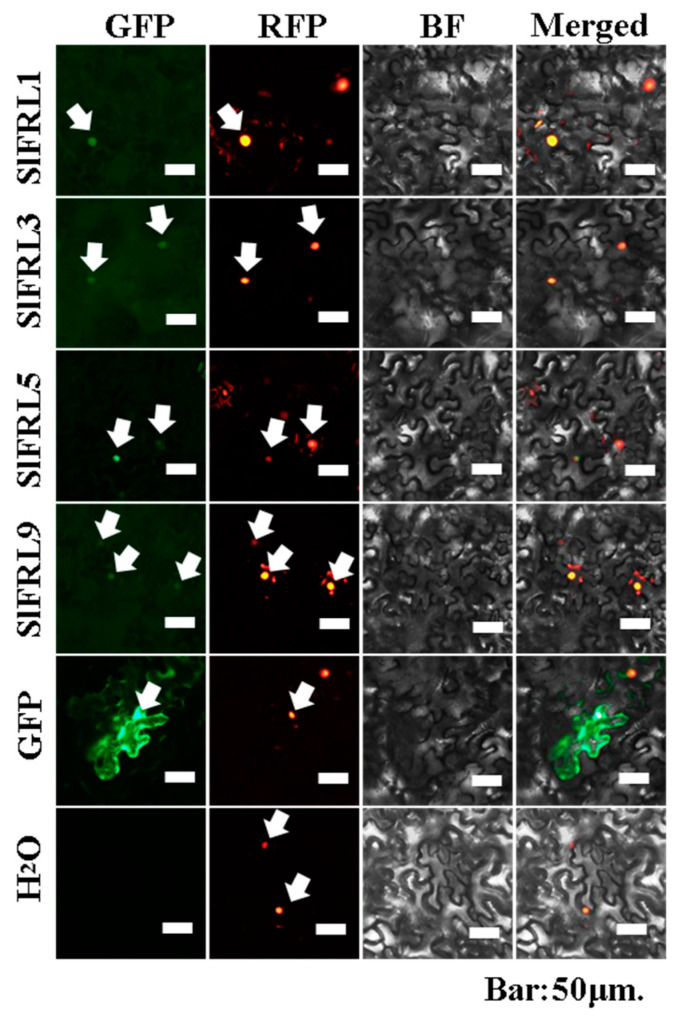
The subcellular localization of SIFRLs, bar = 50μm.

**Figure 6 ijms-23-11264-f006:**
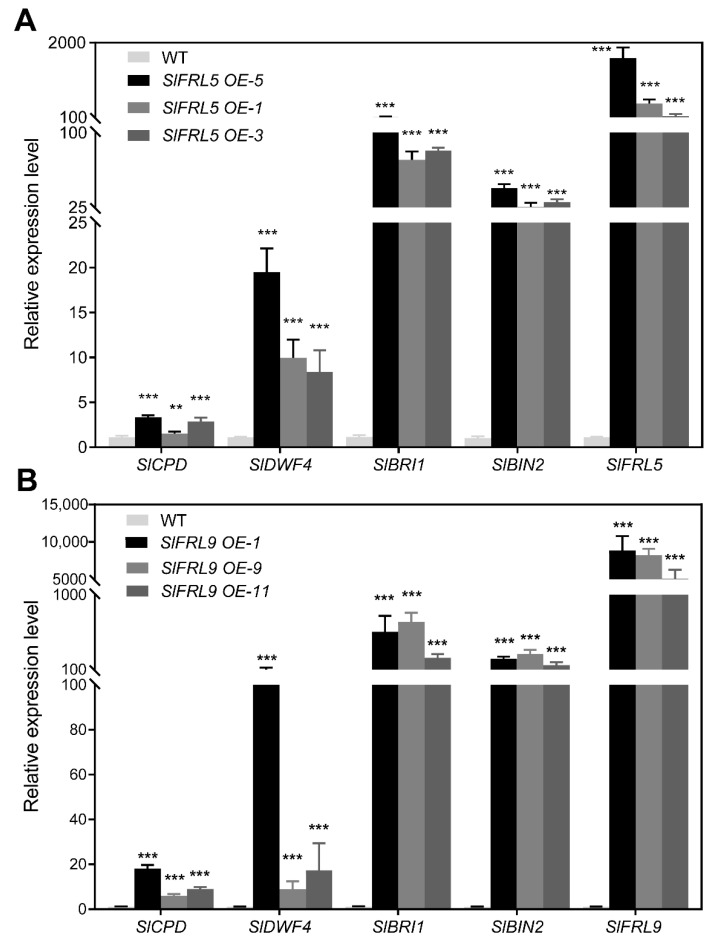
The mRNA levels of different genes in WT and transgenic lines. (**A**,**B**) are the mRNA levels of each gene in *SlFRL5-OE* and *SlFRL9-OE* lines, respectively. *SlUBI3* was used as the internal reference gene, repeated three times and calculated by 2^−ΔΔCT^. In the figure, 2 stars (**) stand for 1% while 3 stars (***) for 0.1% level of significance.

**Figure 7 ijms-23-11264-f007:**
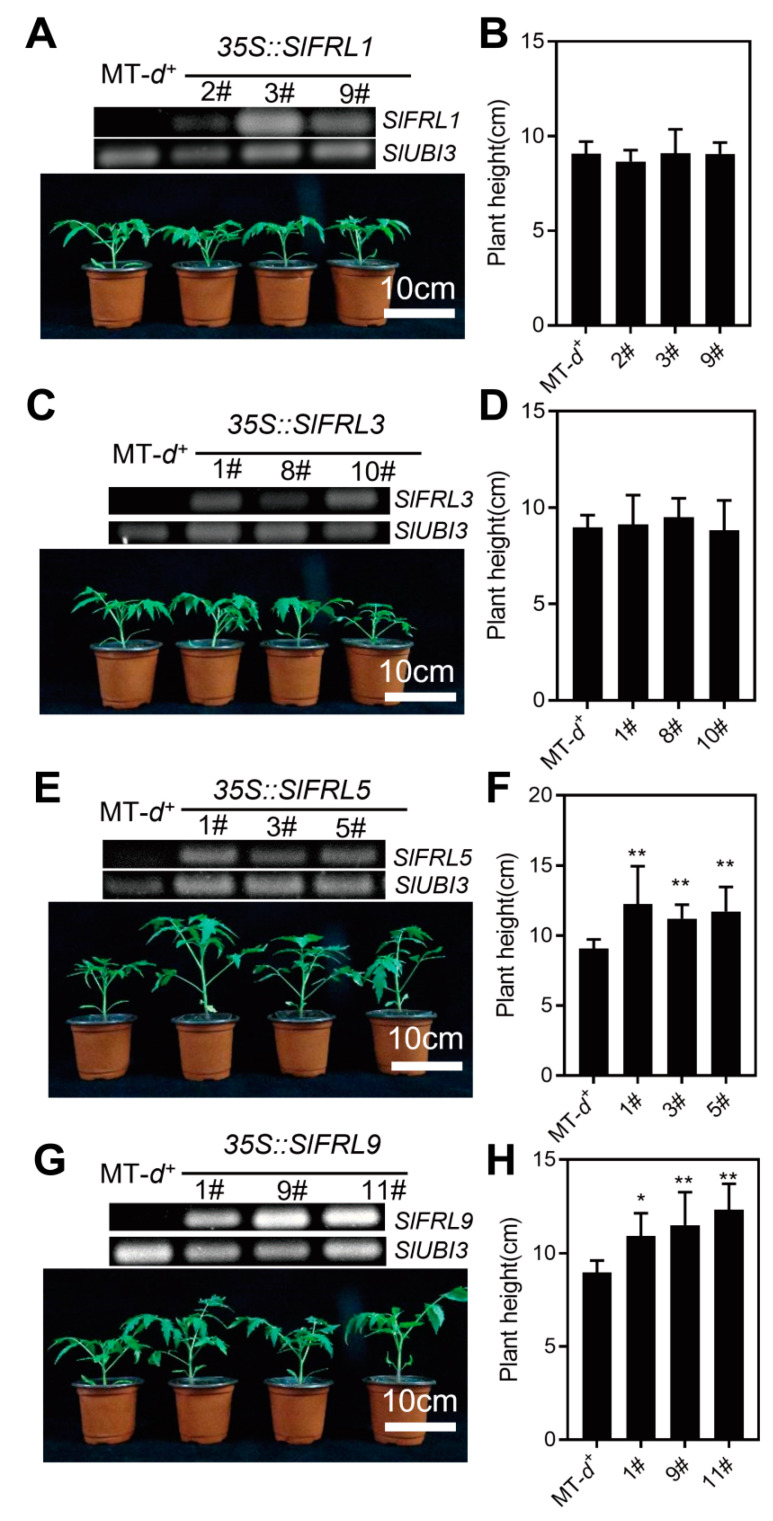
The plant morphology of different overexpression lines in 4 weeks. (**A**,**C**,**E**,**G**) were *SlFRL1-OE*, *SlFRL3-OE*, *SlFRL5-OE* and *SlFRL9-OE* overexpression lines, respectively. The electrophoresis map was used for identification of the materials by RT-PCR. *SlUBI3* was used as the reference gene, and the phenotype of the corresponding transgenic lines for 4 weeks is below. Figs. (**B**,**D**,**F**,**H**) correspond to *SlFRL1-OE*, *SlFRL3-OE*, *SlFRL5-OE*, and *SlFRL9-OE* plant height at 4 weeks, respectively. *n* ≥ 8. In the figure one star (*) stand for 5% level of significance while 2 stars (**) for 1%.

**Figure 8 ijms-23-11264-f008:**
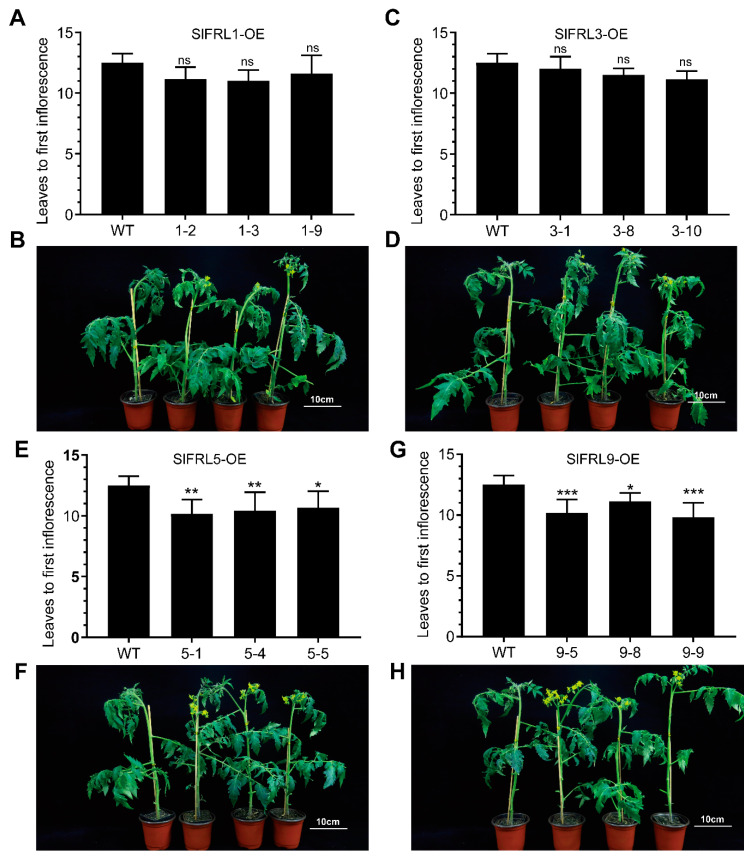
Overexpression *SlFRL5* or *SlFRL9* caused early flowering phenotype in tomato. (**B**,**D**,**F**,**H**) are the flowering pictures of *SlFRL1-OE*, *SlFRL3-OE*, *SlFRL5-OE* and *SlFRL9-OE* lines, respectively. (**A**,**C**,**E**,**G**) are the number of true leaves under the first panicle. *n* ≥ 6. In the figure one star (*) stand for 5% level of significance while 2 stars (**) for 1% and 3 stars (***) for 0.1% level of significance.

**Figure 9 ijms-23-11264-f009:**
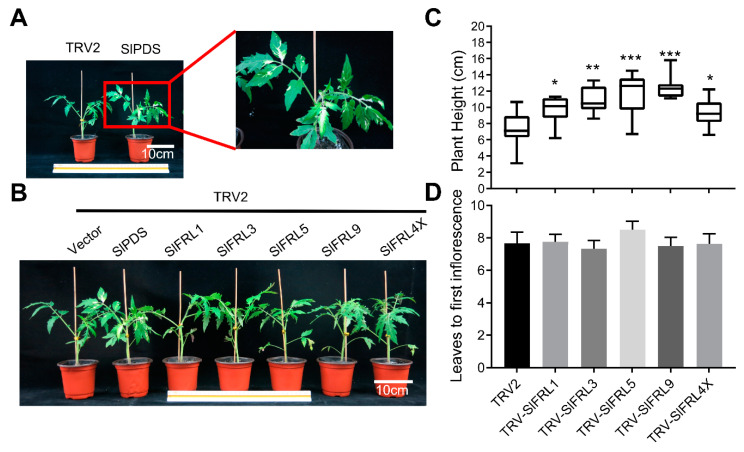
Transient silencing of *SlFRLs* promotes plant height in tomato. (**A**) was the phenotype of TRV2-*SlPDS* plants and their albino leaves, (**B**) was the phenotype of transient transformation materials at 4 weeks, (**C**,**D**) was the plant height at 4 weeks of each material in (**B**) and the number of true leaves under the first panicle. TRV-*SlFRL4X* is a mixture of TRV-*SlFRL1*, TRV-*SlFRL3*, TRV-*SlFRL5* and TRV-*SlFRL9*. *n* ≥ 8. In the figure one star (*) stand for 5% level of significance while 2 stars (**) for 1% and 3 stars (***) for 0.1% level of significance.

**Figure 10 ijms-23-11264-f010:**
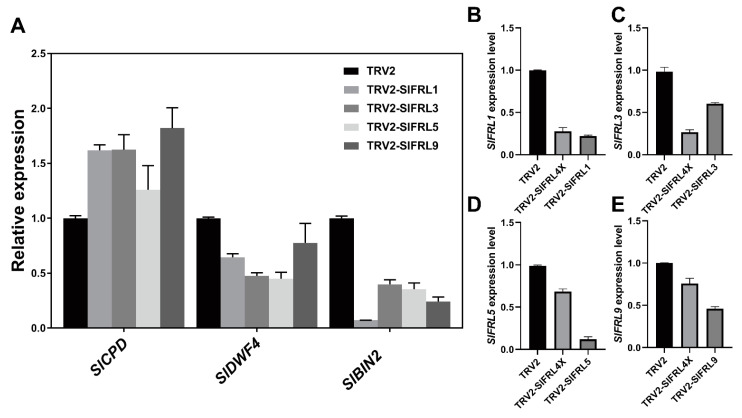
Expression levels of genes after instantaneous silencing of *SlFRLs.* (**A**) was identification of mRNA levels of multiple genes in transient silencing materials, and (**B**–**E**) were identification of mRNA levels of silencing materials. *SlUBI3* was used as the internal reference gene, repeated three times and calculated by 2^−ΔΔCT^.

## Data Availability

The original data involved in the study has been kept in Xiaofeng Wang’s Lab. Accessing to the data can contact to corresponding author Xiaofeng Wang with E-mail: Wangxff99@nwsuaf.edu.cn or first author Bote luo with robert2018@nwafu.edu.cn.

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
