# Peer review of "Brassinosteroid Signaling Downstream Suppressor BIN2 Interacts with SLFRIGIDA-LIKE to Induce Early Flowering in Tomato"

_ijms, 2022, doi:10.3390/ijms231911264_

Round 1

Reviewer 1 Report

Comments to the authors

General comments:

#1 The authors found a total of 12 FRIGIDA-LIKE 15 proteins in tomatoes, all of which included the conserved FRIGIDA domain. First and foremost, the findings of this research provide molecular evidence that BRs regulate the flowering time of tomato plants. This opens the door for additional research into the interactive role that SlFRLs and SlBIN2 play in flowering time, as well as an increased comprehension of the specific pathway that is essential for floral regulation.

#2 The work as a whole has a good scientific quality, however, the author has to focus on polishing the language because some of the phrases lack coherence and logic. Overall, the work is of good quality. In addition to that, I noticed several instances of duplication throughout the write-up. To improve the overall quality of the document, I suggest sending it to an editor who specializes in the English language.

#3 When writing, the use of passive voice rather than active voice is considered to be more appropriate. It was observed that the authors used the pronoun "we" several times throughout the whole body of the manuscript.

#4 It is recommended not to start the sentence with an abbreviation.

#5 When scientific names are being used, such as Solanum lycopersicum, they should be spelt out in their entirety the first time they are used. After that, it is OK to abbreviate the term to S. lycopersicum.

 #6 Because the information about tomato plants, their economic worth, and their relevance was not included until the very last paragraph (line 79), the introduction section could incorporate further information about these topics.

 #7 At the very end of the section under "Introduction," objectives need to be outlined very explicitly.

 #8 Materials and methods should contain and detail all of the experiments. For instance, the authors should describe how they grow tomato plants in the pots, including whether they use seeds or transplants, the age of the transplant, irrigation and other agricultural practices, and so on.

 #9 The results section has a lot of figures; in all, there are 11 figures; figure 10 is even repeated on lines 260 and 264. The sequence of the text should be rearranged based on the number of the figures; for instance, the text pertaining to figure 7 should come before the text pertaining to figure 6; lines 187 and 204, respectively. I would suggest to the authors that they read through the figures and only include the ones that are significant to the conclusion of the study in their work. Providing the readers with a large number of figures will not assist them to achieve the goals.

 #10 Discussion section is well stated; nonetheless, there should be additional tomato-related sources added.

 Other comments:

Line11: kindly, change to “These components are widely studied in Arabidopsis however, very little information is available in tomato.”

 Line 12-15: please rephrase the sentence.

 Line 26: please remove “;”.

 Line 36: Please change to shoot apical meristem.

 Line 36: Please change to A. thaliana.

 Line 93: In Arabidopsis, AtFRL4 regulates early flowering while its overexpressed lines show a late-flowering phenotype [26], which should be discussed more in the discussion section.

Line 111: kindly add tomato before Solanum lycopersicum and use S. lycopersicum instead.

 Line 153: is it YFP or GFP!

 Line 155: Is it total or mRNA? Please review the MM section to ensure compliance.

 Line 156: How the authors decide the young roots. This should be clearly described in MM section. Young and old roots of the 8 weeks old seedlings.

Line 166: the authors mentioned “ previous studies showed…” are these author's studies or other authors, the ref should be cited.

Line 170: H2O is used as a negative control; this should be explained in the MM section as well.

Line 177: Fig. 4 legend standard bar should be explained and moved from Fig. 5 legend.

 Fig. 4 please change mature fruit to red ripe fruits as mentioned in the text. Please use the same scale in the histogram to ensure the comparison. The maximum number in all panels except (C) is 4. All should be changed to 5.

 Fig. 5 please move “bar: 50 µm” to the end of the legend.

 Fig. 5 legend should be modified as it is written mean ± standard deviation (error line) however, it is an image.

 Fig. 6 Plant size, it can be changed to plant morphology. The text of this figure should be mentioned before Fig.7. on the other hand, it should be explained in MM. Is it plant height or shoot length? The scale of B, D, F, and H should be adjusted to 20.

 Fig. 10 was repeated, please change fig. 10 in lines 264 to 11 and modify the text.

Line 354: “Other cultural conditions were same as used followed by reference [43] with some modifications.” What are these modifications?

 Line 367: Please change obtained to obtain.

 Line 369: Please change performed to perform.

 Line 377-378: “For yeast two-hybrid analysis we used the method previously described [43] with some modifications.” What are these modifications?

 Line 411: How do the authors decide on the young roots?

 Line 424: please change “vector used” to vector was used.

 Line 447: Please change combination to combinations.

Line 449: Please change need to needed.

Author Response

Response to Reviewer 1

Dear Reviewer, thank you so much for your constructive suggestions. I have incorporated all the comments and modify this article according to your suggestions. Please have a look and let me know, if there is still any modification need. Your further valuable suggestions, if any, will be highly appreciated.

Point to point response is made in this file. Please have a look at the response, which is bold text format and highlighted with blue color. 

General comments:

#1 The authors found a total of 12 FRIGIDA-LIKE 15 proteins in tomatoes, all of which included the conserved FRIGIDA domain. First and foremost, the findings of this research provide molecular evidence that BRs regulate the flowering time of tomato plants. This opens the door for additional research into the interactive role that SlFRLs and SlBIN2 play in flowering time, as well as an increased comprehension of the specific pathway that is essential for floral regulation.

I appreciate your positive comments.

#2 The work as a whole has a good scientific quality, however, the author has to focus on polishing the language because some of the phrases lack coherence and logic. Overall, the work is of good quality. In addition to that, I noticed several instances of duplication throughout the write-up. To improve the overall quality of the document, I suggest sending it to an editor who specializes in the English language.

The manuscript is completely checked and revised to meet your suggestions.

#3 When writing, the use of passive voice rather than active voice is considered to be more appropriate. It was observed that the authors used the pronoun "we" several times throughout the whole body of the manuscript.

The manuscript is thoroughly checked and changed as suggested.

#4 It is recommended not to start the sentence with an abbreviation.

All the abbreviations at the start of the sentence is changed and replaced with full name.

#5 When scientific names are being used, such as Solanum lycopersicum, they should be spelt out in their entirety the first time they are used. After that, it is OK to abbreviate the term to S. lycopersicum.

Adjusted as per suggestions

#6 Because the information about tomato plants, their economic worth, and their relevance was not included until the very last paragraph (line 79), the introduction section could incorporate further information about these topics.

Relevant information about the tomato plant is added according to the suggestion.

#7 At the very end of the section under "Introduction," objectives need to be outlined very explicitly.

Objectives of the research work is clearly added at the end on introduction

#8 Materials and methods should contain and detail all of the experiments. For instance, the authors should describe how they grow tomato plants in the pots, including whether they use seeds or transplants, the age of the transplant, irrigation and other agricultural practices, and so on.

Material and method section is rewritten as per suggestion and all the cultural practices is outlined.

#9 The results section has a lot of figures; in all, there are 11 figures; figure 10 is even repeated on lines 260 and 264. The sequence of the text should be rearranged based on the number of the figures; for instance, the text pertaining to figure 7 should come before the text pertaining to figure 6; lines 187 and 204, respectively. I would suggest to the authors that they read through the figures and only include the ones that are significant to the conclusion of the study in their work. Providing the readers with a large number of figures will not assist them to achieve the goals.

The results section is fully revised and made according to the reviewer guidelines. Sequence of the text was rearranged and some results and figures are removed as suggested.

#10 Discussion section is well stated; nonetheless, there should be additional tomato-related sources added.

It is added as reviewer suggestion.

Other comments:

Line11: kindly, change to These components are widely studied in Arabidopsis however, very little information is available in tomato.”

Changed according to guidelines of reviewer

Line 12-15: please rephrase the sentence.

Sentences are rephrased and rearranged

Line 26: please remove “;”.

Modify accordingly

Line 26; please change to shoot apical meristem.

Changed as per reviewer suggestion

Line 36: Please change to A. thaliana.

Changed accordingly

Line 93: In Arabidopsis, AtFRL4 regulates early flowering while its overexpressed lines show a late-flowering phenotype [26], which should be discussed more in the discussion section.

Discussed as suggested

Line 111: kindly add tomato before Solanum lycopersicum and use S. lycopersicum instead.

Added as suggested

Line 153: is it YFP or GFP!

It is YFP

Line 155: Is it total or mRNA? Please review the MM section to ensure compliance.

mRNA

Line 156: How the authors decide the young roots. This should be clearly described in MM section. Young and old roots of the 8 weeks old seedlings.

Young and old roots is described in MM section.

Line 166: the authors mentioned “ previous studies showed…” are these author's studies or other authors, the ref should be cited.

It is other’s work and the ref has been cited now.

Line 170: H2O is used as a negative control; this should be explained in the MM section as well.

This point is explained in the MM section as suggested

Line 177: Fig. 4 legend standard bar should be explained and moved from Fig. 5 legend.

Adjusted accordingly

Fig. 6 Plant size, it can be changed to plant morphology. The text of this figure should be mentioned before Fig.7. on the other hand, it should be explained in MM. Is it plant height or shoot length? The scale of B, D, F, and H should be adjusted to 20.

It is adjusted.

Fig. 10 was repeated, please change fig. 10 in lines 264 to 11 and modify the text.

These figures are removed as per suggestions.

Line 354: “Other cultural conditions were same as used followed by reference [43] with some modifications.” What are these modifications?

The modification in the cultural practices was added

Line 367: Please change obtained to obtain.

changed

Line 369: Please change performed to perform.

Changed

Line 377-378: “For yeast two-hybrid analysis we used the method previously described [43] with some modifications.” What are these modifications?

The modifications we made in the experiment is mentioned in MM section

Line 411: How do the authors decide on the young roots?

The plants were uprooted and then roots are separated. Different sections of the roots were made such as mature roots and newly grown delicate roots.

Reviewer 2 Report

The manuscript (MS) submitted by Khan et al. demonstrated presence of FRIGIDA family proteins in tomato, and the interaction between SlFRLs and SlBIN2.1 to regulate early flowering in tomato. The MS reports interesting information, research methodology and outcome of the study have practical application. English minor spell check required. Therefore, I recommend ACCEPTANCE of this manuscript.

Author Response

Dear reviewer: I appreciate your positive comments.

Reviewer 3 Report

The manuscript is written well and is of critical importance to tomato genomics and the breeding research community. Please see the attached review reports for more review inputs. 

Author Response

Response to reviewer 3

Dear reviewer, thank you so much for your constructive suggestions. The article has been thoroughly checked and all the sections are adjusted according to your suggestions. Please have a look and please guide us if any further improvement needed in this manuscript. Your valuable and to the point suggestion encourage me a lot. 

IJMS 1900008 Manuscript Review Report:

Overall Remarks: The manuscript is written well and is of critical importance to tomato genomics and the breeding research community.

Thank you very much for your positive comments.

Abstract: Abstract is bit longer than the recommended word count (200 words), please rewrite the abstract succinctly within the recommended word limit. Also, there are some abbreviations that italicized and some are not so please be consistent throught the abstract. Also used abbreviations should be defined when appears for the first time such as “BR”. Also, sentence should not start with the abbreviation such as “BR signaling ….”

These are very nice suggestions and the abstract has been completely revised on the suggestion.

Introduction: Authors have provided good background of related topics and succinctly provided relevant references. However, last paragraph needs to be revised since it seems more like materials and methods and missing the experimental objective and hypothesis, please make sure to elaborate on research objectives and hypothesis in the last paragraph.

It is revised on the suggestion.

Results: Written well, but it is too wordy. No clear indication of what exactly this result has achieved and very hard to follow.

It is revised on the suggestion.

Discussion: Authors tried to succinctly discuss their findings with earlier published research, but key points seem to have missed. Some discussion section seems to appear like result section and authors are advised to revise the discussion section by highlighting the impact of result findings.

It is revised on the suggestion.

Materials and methods: Overall experiment is designed well and authors have provided details of the methodology used.

Thanks.

Round 2

Reviewer 1 Report

The conserved FRIGIDA domain was present in all 12 of the FRIGIDA-LIKE 15 proteins that the authors discovered in tomatoes. First and foremost, the results of this study offer molecular proof that BRs control the timing of tomato plant flowering. This provides a starting point for further investigation into the interacting function of SlFRLs and SlBIN2 in flowering time as well as a deeper understanding of the particular route required for floral regulation.

However, one item to advise is to write more about tomato plants and place the material up in the introductory part. The work as a whole now has a good scientific quality following revision.